# The Role of CRP POC Testing in the Fight against Antibiotic Overuse in European Primary Care: Recommendations from a European Expert Panel

**DOI:** 10.3390/diagnostics13020320

**Published:** 2023-01-15

**Authors:** Ivan Gentile, Nicola Schiano Moriello, Rogier Hopstaken, Carl Llor, Hasse Melbye, Oliver Senn

**Affiliations:** 1Department of Clinical Medicine and Surgery—Section of Infectious Diseases, University of Naples Federico II, 80131 Naples, Italy; 2Nineth Division of Infectious Disease, Cotugno Hospital, 80131 Naples, Italy; 3Star-shl Diagnostic Centers, 3068 Rotterdam, The Netherlands; 4Department of Public Health and Primary Care, University of Southern Denmark, 5230 Odense, Denmark; 5General Practice Research Unit, Department of Community Medicine, The Arctic University of Norway, 6050 Tromsø, Norway; 6Institute of Primary Care, University of Zurich and University Hospital of Zurich, 8091 Zurich, Switzerland

**Keywords:** C-reactive protein, antimicrobial resistance, primary care

## Abstract

Tackling antibiotic resistance represents one of the major challenges in modern medicine, and limiting antibiotics’ overuse represents the first step in this fight. Most antibiotics are prescribed in primary care settings, and lower respiratory tract infections (LRTIs) are one of the most common indications for their prescription. An expert panel conducted an extensive report on C-reactive protein point-of-care (CRP POC) testing in the evaluation of LRTIs and its usefulness to limit antibiotic prescriptions. The expert panel stated that CRP POC testing is a potentially useful tool to limit antibiotic prescriptions for LRTI in a community setting. CRP POC must be used in conjunction with other strategies such as improved communication skills and the use of other molecular POC testing. Potential barriers to the adoption of CRP POC testing are financial and logistical issues. Moreover, the efficacy in limiting antibiotic prescriptions could be hampered by the fact that, in some countries, patients may gain access to antibiotics even without a prescription. Through the realization of a better reimbursement structure, the inclusion in standardized procedures in local guidelines, and better patient education, CRP point-of-care testing can represent a cornerstone in the fight against antimicrobial resistance.

## 1. Introduction

Inappropriate antibiotic prescribing is a global problem expected to result in a huge increase in deaths and decreased economic output from antibiotic-resistant infections in the next 30 years. Global awareness campaigns [1] and a growing body of research about the consequences of bacterial antimicrobial resistance (AMR) have fostered a consensus among providers that current antibiotic prescribing patterns need to be reduced from their current levels across a range of indications. It has been shown that most antibiotics are prescribed in primary care setting and that acute lower respiratory tract infections (LRTIs) are one of the most common indications for their prescription [2]. Potential drivers of the current rate of antibiotic prescribing include perceived pressure or expectation from patients, fear of complications, low short-term risks (i.e., good tolerability profile for most antibiotics), diagnostic uncertainty, and an underestimation of the serious downstream effects of antibiotic employment by healthcare providers. Evaluation of inflammatory markers such as C-reactive protein (CRP) can help clinicians distinguish non-severe infections from more severe infections in patients with relatively nonspecific LRTI symptoms such as a cough or shortness of breath [3,4,5,6,7,8]. Lower CRP values typically indicate a higher likelihood of mild or self-limiting infections with a good prognosis and, thus, preclude the need for antibiotics [9]. C-reactive protein point-of-care testing (CRP POC) is a well-established diagnostic and prognostic tool that, if used in primary care settings, may have a role in the fight against antibiotic over-prescription in patients with LRTI symptoms. Recent data have shown that CRP POC testing during primary care evaluation for RTI symptoms safely reduces the likelihood of antibiotic prescription [2,8,10,11,12,13,14,15]. In detail, several trials have been conducted to assess the usefulness of CRP POC. A recent metanalysis analyzed those trials with an overall sample of 9444 patients. According to the results, CRP-POCT significantly reduced immediate antibiotic prescribing at the index consultation compared with usual care, especially in the setting of LRTI [12]. However, different methodologies and cut-offs have been used in different trials, indicating the need for common protocols for the use of PCR POC. Despite growing awareness of bacterial AMR and clinical studies that support the use of CRP POC testing as an effective adjunct tool for the evaluation of patients presenting with LRTI symptoms, there is significant variability in uptake and application of this tool across Europe [13]. The purpose of this report is to gain a comprehensive understanding of the factors that influence antibiotic use for LRTIs across Europe, the current role of CRP POC testing in European primary care practices, and the anticipated actions necessary to change antibiotic prescribing practices for LRTIs. 

## 2. Materials and Methods

An extensive project using both quantitative and qualitative survey methods was commissioned to supplement the published literature about CRP POC testing in the evaluation of LRTIs. A leading European market research firm organized an expert panel of six primary care physicians and one hospital physician (The Expert Panel) from a variety of locations and practice settings across Europe (Table 1). These physicians have research interests in the primary care evaluation of LRTIs, the growing risk of bacterial AMR, and the application of CRP POC testing. Countries represented include the United Kingdom, Spain, Germany, Italy, Norway, Switzerland, and the Netherlands, and practice settings included both academic and community-based practices and hospitals. The market research was conducted between November 2021 and February 2022 using multiple quantitative surveys with open response, multiple choice, and Likert scale questions (the meeting schedule and the questions that the Expert Panel were asked are available as Appendix A). The results of the survey were analyzed using descriptive statistics tools. Following on from these results, two thorough and moderated open response forums separated by approximately 6 weeks were recorded and analyzed using qualitative methodology. These interpretation sessions were conducted remotely.

## 3. Results

Several key themes emerged from both the quantitative survey results and the qualitative analysis of the moderated forum, including a collective agreement that current rates of antibiotic prescribing for LRTIs are too high and that CRP POC has an important role in prudently decreasing the prescribing rate. Specific details regarding ideal clinical integration, expected barriers, drivers of change, potential for research, and scope for next steps are summarized in Figure 1 and are further delineated below.

### 3.1. CRP POC Testing as a Pillar in the Fight against Antibiotic Over-Prescription

The Expert Panel estimated that the current rate of antibiotic prescription for LRTIs in European primary care is approximately 2–3 times what is likely required (Figure 2). Multiple clinical studies have proven that CRP POC testing can be effective in the fight against over-prescription of antibiotics, and all physicians on the panel agreed with this position, especially for LRTIs [2,10,12,13,14,16,17]. CRP values can serve as an indicator to suggest an underlying severe infection versus mild or self-limiting infections for LRTIs, and, when combined with their clinical assessment, this test can help a clinician decide if an antibiotic prescription will be beneficial for the patient.

Currently, there is a wide variability in the use of CRP POC testing in the primary care setting across Europe, as illustrated by the significant debate among panel members about the use of CRP POC for patients presenting to primary care for LRTI symptoms. This debate centers around the two most likely uses of CRP POC testing in this setting: a diagnostic aid to accompany clinical assessment, or a prognostic test that informs GPs when deciding to prescribe antibiotics (Table 2). Regardless of one’s utilization preference, all panel members agreed that in order to add diagnostic or prognostic value, the results must truly be point-of-care and rapidly available.

Members of the panel also agreed that CRP testing is preferrable to procalcitonin—another marker that can differentiate severe from self-limiting infections [18,19]. CRP testing is less expensive [20], delivers results more quickly, and has better utility in a lower-acuity setting such as an outpatient practice when compared to procalcitonin. Furthermore, the panel highlighted that procalcitonin is not very well evaluated in the primary care setting. Studies in primary care thus far have found less added value of procalcitonin compared to CRP. The members of the panel agreed that CRP is to be preferred to erythrocyte sedimentation rate (ESR) since the latter is slower to rise and fall in respect to the former. Moreover, ESR could be falsely positive in the case of anomaly of the blood constituents such as monoclonal immunoglobulins. Other blood markers such as transaminases, troponin, or lactate dehydrogenase (LDH) are not specific as markers of inflammatory state and their usefulness in the decision of prescribing antibiotics in the primary care setting is limited.

Quantitative tests must be preferred, as the CRP punctual value is crucial for the test interpretation [21]. In detail, a low-grade elevation of CRP is associated with a wide range of chronic conditions resulting in metabolic stress (e.g., obesity, atherosclerosis, obstructive sleep apnea) and not with acute infection [22,23,24]. Medium- to high-grade elevation of CRP is strongly associated with acute inflammatory state [25,26]. Several factors can cause an acute inflammation such as autoimmune disease, infectious disease, malignant neoplasms, and tissue injury. Medical history and clinical examination are mainstays in the differential diagnosis. Regarding infectious diseases, CRP is elevated in bacterial, fungal, and viral infections However, even considering the limits exposed above, C-reactive protein retains a strong negative predictive value in ruling out the presence of an active bacterial infection, especially LRTI [27]. Semi-quantitative testing may be acceptable if this significantly increases accessibility for CRP POC testing or lowers the cost of adoption, especially in resource-limited settings. 

The Expert Panel felt that CRP POC testing would be best integrated into evaluating and treating patients presenting with LRTI symptoms by conducting a CRP POC test in instances of diagnostic or prognostic uncertainty. If the CRP POC value is high (>100 mg/L), this confirms the need for antibiotics, and for patients with a low CRP measure (<20 mg/L), antibiotics are not needed. However, for patients with borderline CRP POC values or with other concerning features (e.g., age, medical comorbidities), the physician can engage in shared decision-making and perhaps employ delayed prescribing for patients still hesitant to proceed without antibiotics. Potential limits in the role of CRP testing may arise when considering patients receiving immunosuppressive drugs. It has been shown that C-reactive protein levels may not rise in the case of bacterial infection in patients treated with tocilizumab [28], and, given the rising number of molecular-targeted immunosuppressive drugs, it is possible that other drugs may have the same effect. Furthermore, it has been demonstrated that immune disfunction such as the presence of anti-IL-6 autoantibodies may be associated with normal levels of CRP even in presence of a bacterial infection [29]. Physicians should be aware of these limits in cases when the levels of CRP are in contrast with the clinical condition of the patient.

### 3.2. Combining CRP POC Testing with Other Strategies to Reduce Antibiotic Over-Prescription

Though all panel members agreed that CRP POC testing plays an important role in fighting antibiotic over-prescription for LRTIs, they also concurred that it should not be used as athe only method to reduce antibiotic over-prescription. Instead, its greatest impact could be achieved when it is used in combination with other strategies, as seen in Figure 3. All the strategies discussed can be seen in Table 3. 

#### 3.2.1. Communication Skills

One such strategy is employing specific communication skills and tools aimed at reducing antibiotic prescribing. Communication skills are a foundational skill in medical education, but specific training around antibiotic prescribing and CRP testing is likely not a widespread part of the clinical curriculum of medical doctors, particularly for those who completed medical training prior to the increased awareness of bacterial AMR as a global issue. Communication skills must also include other aspects not necessarily specific to bacterial AMR or POC testing, such as exploring worries, expressing empathy for bothersome symptoms, sharing information about expectations for a patient’s disease course, and discussing appropriate return precautions so that the patient feels heard and that the general practitioner (GP) is taking his or her symptoms seriously.

Unfortunately, the success of these tools varies based on practice time and personnel constraints. Furthermore, in countries such as Italy and Spain, there is a strong cultural belief about the role of antibiotics, and it seems unlikely that one short appointment could change a patient’s longstanding belief. Enhanced consultation skills around discussions of antibiotic indications and bacterial AMR are needed, despite the fear that consultations will take longer, but as patients become more informed on the topic over time, it is possible that consultations will ultimately take less time in the long run. Panel members agreed that CRP POC testing could be integrated into communication strategies, perhaps as part of a decision aid, because it is quick and individualized to each patient. Anecdotally, panel members reported that patients generally accept CRP POC testing and agreed that low CRP values could also be a springboard for patient education in countries such as Italy or Spain where patients are misinformed on indications for antibiotic prescriptions. CRP POC testing could also improve clinic flow by shortening these conversations about the need for antibiotics, because, in some instances, these conversations can take more time than the POC testing itself. Other communication strategies could include global campaigns aimed at increasing awareness of antimicrobial resistance or individual patient stories whose lives have been personally affected by antimicrobial-resistant infections. However, these strategies are less personalized and will likely need to be tailored to the beliefs of a country’s culture.

#### 3.2.2. Delayed Prescribing and Deprescribing

Delayed prescribing is another commonly employed tactic in reducing antibiotic prescriptions [30]. This strategy can work particularly well when the doctor or patient feels unsure or unsafe not using antibiotics, and data show that patients who heed their physician’s advice and forgo taking antibiotics feel more confident without a prescription the next time they have similar symptoms [31]. Moreover, delayed prescribing requires follow-up communication between the patient and the physician to ensure clinical improvement so that the physician can reassess the patient’s need for antibiotics if they have ongoing symptoms. Panel members expressed concern that this follow-up can negatively impact their workflow. Some were skeptical about patients truly delaying taking antibiotics and assumed that most patients fill in the prescription despite the advice to wait. In countries where awareness of bacterial AMR is low and patients’ expectance for antibiotics is high, delayed prescribing was generally felt not to be an ideal strategy, as the risk is high that patients will take antibiotics in any case. Many panel members felt that delayed prescribing needs to be combined with other strategies to truly be successful in reducing antibiotic use, such as more intensive communication with the patient or using information tools. One way to incorporate delayed prescribing with CRP POC testing could be by using the borderline CRP value (>20 mg/L but <100 mg/L) to provide prognostic information to patients that supports their physicians’ advice that antibiotics are not needed. 

Related to delayed prescribing, in countries with over-the-counter access to antibiotics, patients with LRTI symptoms may initiate an antibiotic course themselves without physician input and present later in their disease course for their physician’s guidance in the discontinuation of their antibiotic course, known as “antibiotic deprescribing” [32]. CRP POC testing can be a valuable tool in assuring these patients or as a way to convince them to discontinue their antibiotics. However, the Expert Panel felt that CRP POC testing has less of a role in monitoring antibiotic efficacy over time or in the decision to terminate a physician-prescribed antibiotic course early as compared to its use in deciding whether to prescribe or not at the outset. 

#### 3.2.3. Molecular Diagnostic Testing

Several panel members felt that combining CRP POC testing with diagnostic tests for SARS-CoV-2, influenza, or strep A testing could be valuable, especially if these molecular tests could also deliver POC results. For example, in the context of COVID-19, CRP POC testing can help to evaluate disease severity [33,34,35,36] or, together with clinical evaluation, suggest the need for hospitalization [37,38,39,40].

### 3.3. CRP POC Testing in the COVID Era 

COVID-19 has become one of the most common LRTIs in the world since it was declared a global pandemic by the WHO in March 2020 [41]. The panel evaluated potential applications of CRP POC testing specifically related to COVID-19. The consensus among panel members was that low CRP values still have value in precluding the need for antibiotics for patients presenting with LRTI symptoms in the COVID era, but in the patients with higher CRP values, it may be challenging for physicians to determine the need for antibiotics without also knowing the patient’s COVID status. Many agreed that CRP POC testing could help to evaluate disease severity in patients who are known COVID-positive, potentially ruling out a superimposed bacterial infection or indicating the need for hospitalization regardless of the underlying etiology. That being said, the consensus was that other factors including oxygen saturation, vaccination status, and medical comorbidities would likely be prioritized over CRP levels in the decision for hospital referral, as the literature suggests [37]. 

### 3.4. Barriers to Adopting CRP POC Testing

Though the experts agreed that CRP POC testing has an important role in reducing antibiotic prescriptions for LRTIs and identified many avenues for its use in everyday primary care, the panel members identified key barriers to increased utilization in their countries (Figure 4). 

#### 3.4.1. Financial Structure

Several panel members pointed to the upfront cost and reimbursement structure for POC testing as a major barrier. For example, in some parts of Europe, the cost of POC testing devices falls on individual GP practices unless it is adopted by the entire region or country. In these cases, physicians are not reimbursed for testing, unintentionally creating a negative incentive for GPs to adopt testing. Furthermore, if the reimbursement structure is overly complex or increases bureaucratic burdens for the practice, the likelihood of use is decreased. 

#### 3.4.2. Logistics

In some countries, GPs have limited time per visit or limited ancillary support to run tests, so they may not be able to engage in shared decision-making over the need for antibiotic prescription or to even run the test on top of the rest of their duties during a 10 min office visit. 

#### 3.4.3. Patient Access to Antibiotics

Another complicating factor is patient access to antibiotics. In Spain, antibiotics are available over-the-counter, and other patients across Europe may have leftover antibiotics from previous courses, so CRP POC testing performed in the GP practice may not prevent patients from taking antibiotics despite physician recommendation that they are unnecessary, meaning health systems may not see this technology as beneficial. 

#### 3.4.4. Implementation Strategy

Lastly, some countries lack a clear implementation strategy for incorporating CRP POC testing into the care of patients with LRTIs. To encourage appropriate testing and treatment, GPs need guidelines to indicate where, when, and how testing should be used. Experts across Europe identified the incorporation of CRP POC testing into guidelines as the most important leverage point for increasing usage, but there is significant variability among national and international guidelines about the role of CRP POC in the evaluation of patients with LRTIs. Currently, some countries make no mention of CRP POC testing in their national guidelines, while others encourage its use despite limited access to the technology [13]. Furthermore, some panel members felt that their national guideline recommendations were reactive to the lack of availability of testing technology locally, rather than prescriptively encouraging practitioners or national governing bodies to increase access to and utilization of testing.

### 3.5. CRP POC Reimbursement—Current Landscape and Moving Forward

Uptake of new technologies in healthcare is often driven by underlying financial structures, and the panel discussed how financial support and reimbursement in their respective countries drove local access and use of CRP POC testing (Figure 5). In Spain, Italy, and the UK, CRP POC testing is not used in the decision to treat LRTIs with antibiotics because GPs are not reimbursed for testing and POC machines must be paid for and maintained by GPs. In Switzerland, Norway, and the Netherlands, GPs regularly perform CRP POC testing because POC testing is reimbursed. Typically, patients either wait an additional 10 min for test results or are dismissed and called later if necessary. In these countries, the testing devices are purchased and maintained by the GPs—except for the Netherlands, where the lab purchases the POC machine—but the reimbursement for testing costs combined with the clinical utility makes the initial investment worthwhile. Germany has low reimbursement for POC testing and currently only covers qualitative and semi-quantitative testing (i.e., tests with negative/positive results or with cut-offs but no discriminate value), but this may change with the rise of POC testing technologies.

Potential incentive structure changes to encourage the use of CRP POC testing could include monitoring rates of both antibiotic prescriptions and CRP POC testing usage to provide real-time feedback to over-prescribers and/or under-testers and to tie reimbursement to these data. In essence, those who over-prescribe or those who fail to appropriately test prior to prescribing antibiotics may receive a lower reimbursement than those who are taking the recommended measures. However, for any new incentive system to be successful, panel members agreed that good-quality metrics are needed for monitoring, rather than relying on crude measures such as the number of tests performed or the number of antibiotic prescriptions. This approach would require significant checks and balances for health system administrators or healthcare payors because this could put GPs in a challenging position that may lead to a change in GP behavior, with money as an incentive rather than clinical reasoning. Health system administrators also need to be sensitive to the fact that economic incentives for testing alone may lead to over-testing, which increases the likelihood of spurious results that lead to unnecessary workups. Conversely, over-testing may have less overall risk and long-term cost than over-prescribing antibiotics. Furthermore, the panel felt that if CRP POC testing becomes mandatory even in clear-cut cases requiring antibiotic prescription (e.g., radiographically confirmed pneumonia, purulent COPD exacerbation), there would likely be backlash from providers over a perceived lack of trust in their clinical assessment. One point brought forth for discussion among the panel was the potential utility for combined POC testing platforms, with one machine performing multiple POC tests such as CRP, rapid influenza/strep, and white blood cell count. A combined test platform may be more useful in overcoming economic objections as the versatility adds value that may be better appreciated by administrators and other stakeholders.

### 3.6. CRP POC Testing and the Role of Clinical Guidelines 

Despite agreement on the need for guidelines that incorporate CRP POC testing, adherence to guidelines was felt to be variable across the board, with variations attributed to a reluctance to change ingrained habits. However, younger generations of GPs may have better uptake of guidelines because they were trained in a guideline-driven healthcare climate, whereas the older generations of GPs were not. GPs are also in a challenging position compared to their specialist colleagues because they must stay up-to-date on guidelines across many disciplines. As such, it can be hard to keep up, especially when guidelines change frequently. Depending on the geographic location, GPs may be more likely to follow the recommendations of local over EU-wide guidelines, which, at times, are at odds with the recommendations of local guidelines. For example, in the UK, guidelines from the National Institute for Health and Care Excellence (NICE) are typically followed very closely, regardless of their agreement with EU guidelines on a given issue or condition. In any case, integration of CRP POC testing into clinical guidelines will be essential to increase the utilization of this technology in the primary care setting in an effort to optimize antibiotic use (Figure 5). Key players to achieve this will be local, national, and international GP organizations; multidisciplinary physician groups; and local and national health authorities alongside strategic partners in research, industry, and insurance.

### 3.7. Anticipated Drivers of Change

Looking to the future of CRP POC testing and the fight against bacterial AMR, one of the primary driving forces to reduce antibiotic over-prescription for LRTIs will be patient attitudes, and knowledge about antimicrobial resistance and the role—or lack thereof—of antibiotics in treating LRTIs. Some panel members suspect that the current SARS-CoV-2 pandemic could help improve health literacy, particularly as relates to respiratory disease and POC testing, because the general public has been flooded with information from the media about diagnostic testing and treatment. Another important change will be the integration of specific communication skills and decision aids into medical training. Though this will take years to exert its full effect, it will prepare the next generation of doctors for success in addressing antimicrobial resistance. Lastly, as technology improves, more rapid and accurate diagnostic testing and the integration of data into digital dashboards can help guide our decision-making in a real-time manner. By combining diagnostic information at the time of initial consultation with real-time clinical data on local prescribing and resistance patterns, GPs could provide the right treatment for the right patient in a timely manner.

### 3.8. Potential for Research or Public Health Projects

As access to CRP POC testing expands, it will be important to better understand its role in reducing bacterial AMR and relate this knowledge to physicians and other allied health professionals along with the general public. One potential area of research would be to combine molecular diagnostic tools with CRP testing to understand how CRP levels respond to various sources of infection. On a broader scale, another important project would be further understanding if widely available CRP POC testing impacts antibiotic prescribing and how this impacts patient understanding and expectations.

Practically speaking, research on CRP POC testing should also be translated into the day-to-day life of a GP. This information may come in various forms, including decision aids that incorporate both antibiotic prescribing and CRP POC testing, continued education for GPs on POC testing and the aforementioned communication skills to decrease antibiotic prescribing, and high-quality, culturally specific evidence that these strategies have a measurable positive impact on prescribing quality, bacterial AMR, and local healthcare economics. Countries could consider involving pharmacists as additional stakeholders, particularly where antibiotics are accessible over-the-counter, but this must be in collaboration with physician colleagues and within their scope of training in order for this to be successful. Most importantly, this research needs to be broadcast as education and health literacy campaigns to fight back against the perception that antibiotics are always needed to treat LRTIs. 

### 3.9. Think Global, Act Local

Moving forward, the panel agreed starting locally may be the most successful entry point to increase the usage of CRP POC testing. Local guidelines and culture have the greatest influence on both physician behavior and patient expectations, so working at this level will likely be more fruitful than trying to start Europe-wide. Local insurance companies (where applicable) can be a good place to establish a foothold into the market, because if one payer takes a chance and shows success, others will probably follow. Working with local GPs or multidisciplinary groups to influence guidelines will also not only raise awareness of CRP POC testing in LRTIs, but hopefully spur reimbursement at a broader level to ultimately improve access. Regardless of the starting point, it will be important not to simply focus on developed countries such as those in the EU, but also to take this technology to resource-limited countries who are also contributing to the problem of antibiotic over-utilization. 

## 4. Conclusions

Bacterial AMR is a global problem with widespread ramifications that is driven by misuse and abuse of antibiotics, with at least one out of every two prescriptions for antibiotics being unnecessary. Though new medicines can be developed to combat resistant bacterial strains, moving forward without changing the underlying problem of antibiotic overprescribing is not only postponing the fight against bacterial AMR, but may also complicate the future fight as more resistant strains are developed and the robustness of resistance is enhanced. 

While bacterial AMR is a very challenging problem to address, there is consensus on the best first steps and expected success from those actions. Fundamentally, bacterial AMR needs to be addressed with a widespread stepwise approach, employing strategies such as education on the scope of the problem, communication skills to translate this issue to patients, and including new technologies such as CRP POC testing to limit antibiotic use. 

The true challenge in addressing bacterial AMR comes not in formulating strategies, but rather implementing them in everyday practice. The breadth and depth of stakeholder engagement required to impart change locally, nationally, and internationally is daunting; physicians, patients, policy makers, physician societies, and industry representatives all need to be engaged to implement the required actions. To say the least, an energetic and comprehensive long-term coordination effort is essential. For the sake of analysis and debate, a three-phased approach can be imagined, with phase I consisting of local development guidelines shaped by a well-respected and influential international team of experts, covering the following:Updated recommendations for the use of CRP POC testing for LRTIs and the prescribing of antibiotics;Development of a straightforward implementation strategy guiding the use of CRP POC testing in general practice, specifically accounting for local culture and barriers;Recommendations for the monitoring of appropriate CRP POC testing and antibiotic prescribing;Development of culture-specific educational and communication tools;Definition of further research needs.

This local guidance then lays the foundation for the development of regional implementation plans in the second phase, while the third phase turns to focused monitoring and global expansion. It is only through an effort like this that true progress can be made in the fight against bacterial AMR.

## Figures and Tables

**Figure 1 diagnostics-13-00320-f001:**
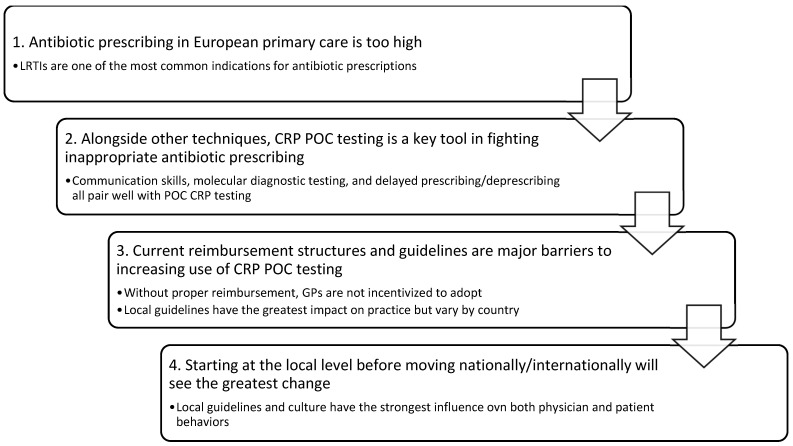
Summary of key findings from the Expert Panel regarding the utility of CRP POC testing in LRTIs.

**Figure 2 diagnostics-13-00320-f002:**
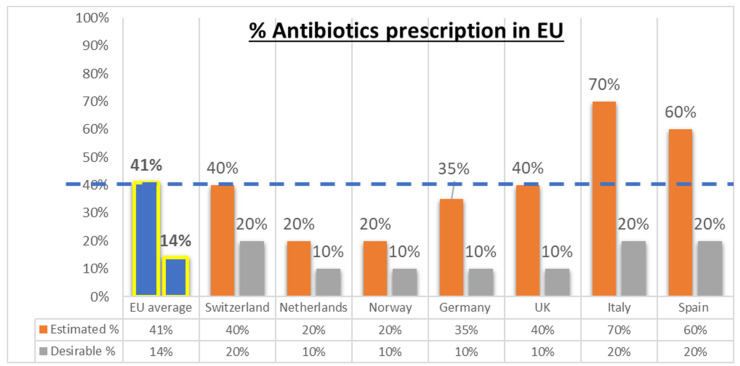
Perceptions of antibiotic prescribing rates for LRTIs in Europe by country. Q: In your country, do you know the approximate percent of antibiotic prescriptions for Lower respiratory tract infections in primary care and what percent would be desirable from your point of view? The dotted blue line represents UE average.

**Figure 3 diagnostics-13-00320-f003:**
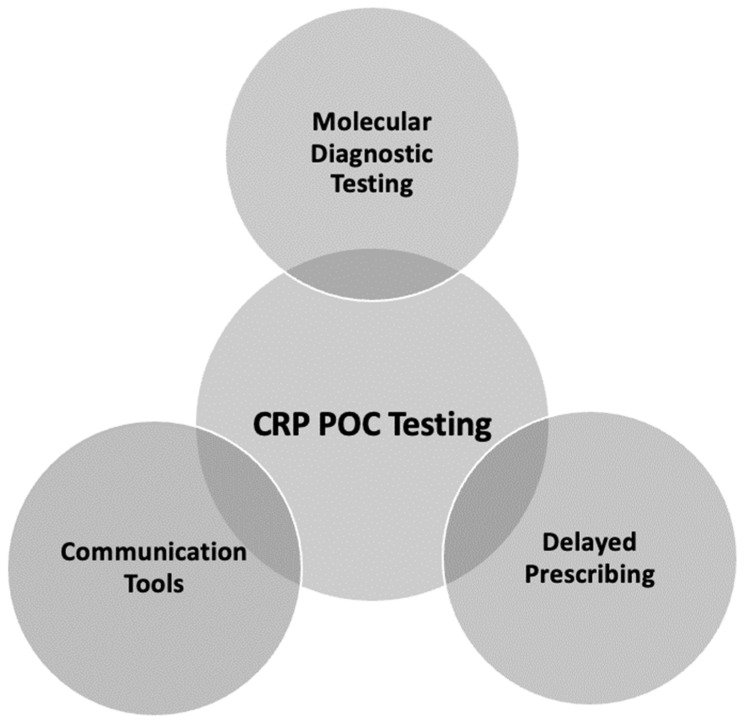
Strategies to reduce antibiotic over-prescription.

**Figure 4 diagnostics-13-00320-f004:**
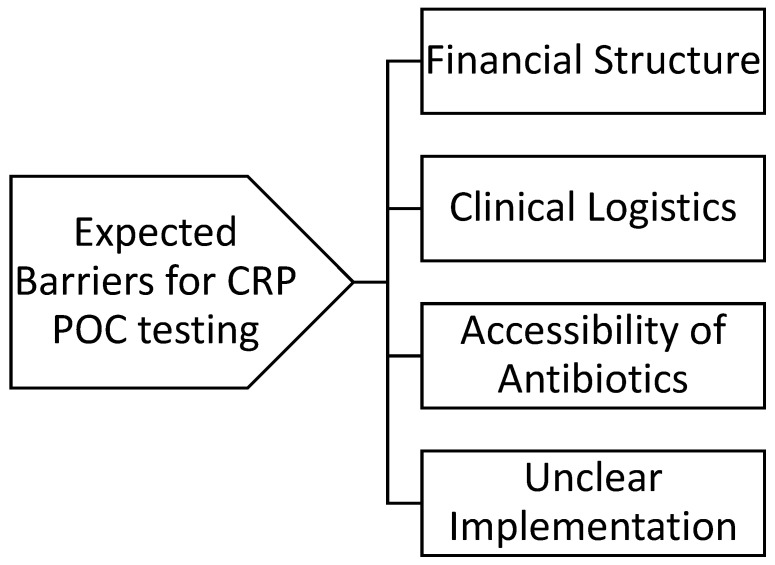
Expected barriers for CRP POC testing.

**Figure 5 diagnostics-13-00320-f005:**
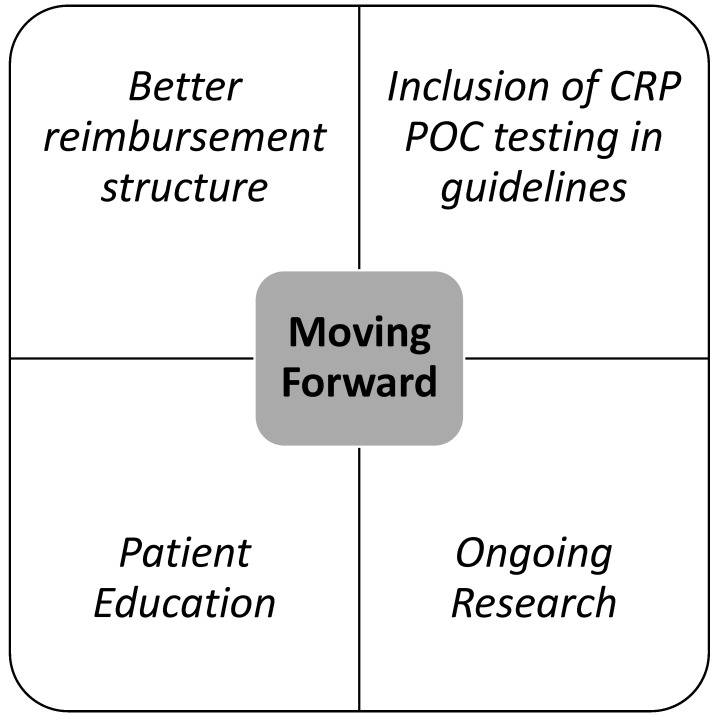
Key elements of reducing inappropriate antibiotic use moving forward.

**Table 1 diagnostics-13-00320-t001:** The Expert Panel.

Attila ALTINER	GERMANY	Professor at the Institut für Allgemeinmedizin and Head of the Institute of General Practice of the University of Rostock
Nick FRANCIS	UK	Professor of Primary Care Research at the University of Southampton/School of Medicine, Neuadd Meirionnydd, University Hospital of Wales
Carl LLOR	SPAIN	General practitioner at the Primary Healthcare Centre, Via Roma, Barcelona, Spain, and Associate Professor at the Department of Public Health and Primary Care at the University of Southern Denmark, Odense
Ivan GENTILE	ITALY	Professor of Infectious Diseases, Chief of the Infectious Diseases Unit AOU Federico II, Director of the Residency Program of Infectious & Tropical Diseases University of Naples Federico II, Naples, Italy
Hasse MELBYE	NORWAY	Professor of General Practice, General Practice Research Unit, Department of Community Medicine, The Arctic University of Norway
Oliver SENN	SWITZERLAND	Professor at the Institute of Primary Care, University of Zurich and University Hospital of Zurich
Rogier HOPSTAKEN	NETHERLANDS	General practitioner and innovation specialist at Star-shl diagnostic centers and chair of the special interest group for POCT of the World Organization of Family Doctors (WONCA)

**Table 2 diagnostics-13-00320-t002:** The current primary care debate about CRP POC test utilization.

Current CRP POC Test Use:	Advantages	Disadvantages
Diagnostic uncertainty following clinical assessment	May be most efficient use when combined with education and communication training	Only influences decisions for patients who present diagnostic dilemma Does not provide definitive diagnostic answer like a molecular diagnostic test would Majority of physicians are confident in their clinical assessment, so will have limited impact in cases where physicians feel certain
Confirmatory test after decision to prescribe	Less reliant on clinical judgement—a simpler message Potentially large influence on prescribing patterns, particularly in high-prescribing countries Potentially improves physicians’ overall diagnostic accuracy through learning from additional information Easy to communicate to patient	Potentially redundant and unnecessary for high-confidence diagnoses, particularly in low-prescribing countries Obligatory testing may lead to physician perception that clinical assessment is not valued Physicians may feel undue pressure to conduct a CRP test Physician acceptance of a confirmatory test is challenging in some cultures/countries Patient acceptance may be difficult in countries where antibiotics are available without prescription

**Table 3 diagnostics-13-00320-t003:** Strategies to reduce antibiotic prescribing in LRTIs.

Quantitative CRP POCT	Clinical assessment remains the primary decision-driver in antibiotic prescription (and its reduction) in LRTIs Supports differentiation of viral and self-limiting bacterial infections from severe bacterial infections in patients presenting with symptoms of LRTI Proven add-on to reduce antibiotic prescribing in LRTIs
Semi-quantitative CRP POCT	Only useful if more than 1–2 cut-offs are provided, e.g., 20–40-100 mg/L Data from studies with quantitative CRP POC tests are not generally transferrable to semi-quantitative tests Resource-limited settings could benefit because of lower costs and increased availability
Delayed prescribing	Not optimal as antibiotics are either needed at the time point of the patient visit or not May be useful for specific patients Should be combined with improved communication Not useful in countries where awareness of bacterial AMR is low and patients’ expectance for antibiotics is high
Communication training and tools	Impactful communication is key, but takes time Needs to be supported by communication tools (online information, patient leaflets, awareness campaigns) Training on communication techniques can be beneficial but should not be prescriptive
Procalcitonin (POCT)	Not well evaluated and not sensitive in the primary care setting More expensive than CRP testing Not generally available
Influenza A and B POCT	Should be used only during the influenza season in selective patients
Strep A POCT	Should be used only when in doubt Useful in countries where prescribing for upper RTIs is high

## Data Availability

Not applicable.

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
