# Peer review of "The Role of CRP POC Testing in the Fight against Antibiotic Overuse in European Primary Care: Recommendations from a European Expert Panel"

_diagnostics, 2023, doi:10.3390/diagnostics13020320_

Round 1

Reviewer 1 Report

The authors reported the expert panel report on C-reactive protein point of care (CRP POC) testing in the evaluation of LRTIs and its usefulness to limit antibiotic prescriptions. They reported CRP POC testing is extremely useful to limit antibiotic prescriptions for LRTI in a community setting. Further, they proposed that CRP POC must be used in conjunction with other strategies like improved communication skills and the use of other molecular POC testing.

The work provides an interesting idea in the diagnosis and prevention of abuse of antibiotics. As a reader, I have some intriguing questions and few major concerns.

1.       The authors should provide the data of approximately 100 or more patients where CRP POC testing was applied and their results.

2.       The CRP level can also increase in viral, fungal infections and in some other cases. For example, a laboratory reference value for a healthy man should by < 6.0 mg/L. In case of arthritis the CRP also increased. What recommendations can be made here?

3.       Will the users be satisfied with CRP POC testing ?

4.       Can the authors provide some data of CRP POC testing in cases of bacterial, viral, fungal, autoimmune and other inflammatory diseases? How it will differentiate and help in prescription of antibiotics in these cases?

5.       The work is mostly descriptive and based on opinions of panel. It will be good to add where CRP POC testing failed.

6.       Lines 23-24 “The expert panel stated that CRP POC testing is a extremely useful to limit antibiotic prescriptions for LRTI in a community setting.” This statement lacks the supporting data in results. Better to change the wording of sentence.

7.       Lines 95-98. CRP is not the only test that can help clinician decide if an antibiotic prescription will be beneficial for the patient. In comparison to other blood tests such as ALT, ESR, Troponin, ck, ldh etc. how the authors think the CRP POC testing is better. 

Reviewer 2 Report

The paper is well-informed on the problem of antibiotic overprescribing. The authors suggested that the CRP POC should be performed to reduce overprescribing. This manuscript is more like a review than an original article paper. The method is the survey method. It might be better to provide the survey questions as a supplement. The statistics application in the study should be mentioned.

Round 2

Reviewer 1 Report

The article seems fine and excellent. 

Reviewer 2 Report

The authors already changed and add more information including the supplement as well.